

# A novel 3D LiDAR deep learning approach for uncrewed vehicle odometry

Wang QiXin and  Wang Mingju

Information Department, Shiyan Taihe Hospital (Affiliated Hospital of Hubei Medical College), Shiyan, HuBei Province, China

## ABSTRACT

Self-localization and pose registration are required for sound operation of next generation autonomous vehicles under uncertain environments. Thus, precise localization and mapping are crucial tasks in odometry, planning and other downstream processing. In order to reduce information loss in preprocessing, we propose leveraging LiDAR-based localization and mapping (LOAM) with point cloud-based deep learning instead of convolutional neural network (CNN) based methods that require cylindrical projection. The normal distribution transform (NDT) algorithm is then used to refine the former coarse pose estimation from the deep learning model. The results demonstrate that the proposed method is comparable in performance to recent benchmark studies. We also explore the possibility of using Product Quantization to improve NDT internal neighborhood searching by using high-level features as fingerprints.

# INTRODUCTION

There has been a growing interesting in the study of autonomous vehicles in recent years (*Yoneda et al., 2014*). Autonomous vehicles are seen as an emergent technology that could help manage traffic efficiently and create new business opportunities amidst the challenges of road safety, law and governance (*Bagloee et al., 2016*). The role and control of autonomous vehicle are dependent on the levels of automation defined by the Society of Automotive Engineers (SAE). There are six levels of control from no automation (0) to full automation (5). For autonomous vehicles to operate safely, they need to be able to sense, plan and act continuously in their environment (*Ilci & Toth, 2020*). Safety is thus of utmost importance in automated driving. *Reid et al. (2019)* highlighted the importance of localization where precise positioning is required for enabling an autonomous vehicle to remain within its lane so that it can operate and navigate safely in various environments.

To mitigate the risks of road crashes, autonomous vehicles rely on a combination of sensors to identify their current position and orientation (*Meng, Wang & Liu, 2017*) as accurately as possible. While Global Navigation Satellite System (GNSS) has been popular for providing localization details, loss of sensor data arising from signal blockage or interference in Global Positioning System (GPS) means that the usability of GNSS is limited to only clear sight environments (*Jeong et al., 2019*). In cases where GPS malfunctions, autonomous vehicles need backup sensors of optimal modality to retain reliability of

Corresponding author
Wang Mingju, 593961345@qq.com

self-localization and/or further simultaneous localization and mapping (SLAM) (*Hemann, Singh & Kaess, 2016*; *Sumikura, Shibuya & Sakurada, 2019*), which can be used to estimate vehicle trajectory given a set of poses (*Gomez-Ojeda et al., 2019*). LiDAR is among the most common sensors deployed in autonomous vehicles (*Lohani & Ghosh, 2017*) because of its superior physical properties. LiDAR measures the distance to its origin using the laser ranging technique. Reflection points with certain reflection intensity level and angles compose an omni-directional point cloud scan.

Many traditional machine learning algorithms have been proposed for LiDAR-based registration, including prevalent point-based methods like iterative closest point (ICP) and normal distribution transform (NDT). In general, ICP is slow and not applicable in SLAM because of its large computation time (*Ilci & Toth, 2020*). Because pre-establishing correspondences is not required in NDT, it is robust against sensor noise (*Sobreira et al., 2019*). Although NDT is also sensitive to the initial estimate, it is faster than ICP because it performs point-to-distribution instead of point-to-point registration (*Ilci & Toth, 2020*). While feature-based methods require huge computations, they are less affected by missing spatial points in subsequent scans (*Li et al., 2019a*) and are able to operate in real-time with high accuracy in pose estimation (*Li et al., 2020*). Deep learning methods have exhibited good performance in recent years (*Li et al., 2019a*) against traditional machine learning methods that require hand-crafted features (*Li et al., 2020*). Deep learning methods have also recently been applied on LiDAR scans, especially in coarse pose estimation and place recognition (*Li et al., 2019b*; *Yin et al., 2018*; *Schaupp et al., 2019*).

## Dedicated contributions

The goal of the proposed solution is to reduce the number of iterations required for NDT by introducing rough guesses. While there are limitations to each scan matching method, the proposed solution combines both feature-based and distribution-based methods for their complementary strengths. This article proposes employing deep learning on point cloud data to estimate pose and subsequently use that estimate as the initial value in NDT for scan matching. Figure 1 shows the general architecture of the proposed solution. The novelty of this solution lies in its ability to optimize the time required to search for an initial optimal value to be used in NDT, thereby achieving higher localization accuracy. This solution does not require 3D to 2D image projection for point cloud scans, preventing information loss during dimension reduction.

## Overall architecture representation

The remaining of this report is arranged as follows: 'Related Work' introduces the related works in scan registration for point- based, distribution-based, feature-based and point cloud based feature learning methods. 'Methods' describes our proposed approaches as well as the odometry system architecture for runtime testing on the approach. 'Experiments' describes the experimental setup and reveals the outcomes of the experiments. 'Reflections' incorporates our reflections on the experimental results. It also highlights the limitations, project enhancement and future works. 'Conclusion' provides the conclusion of this article.

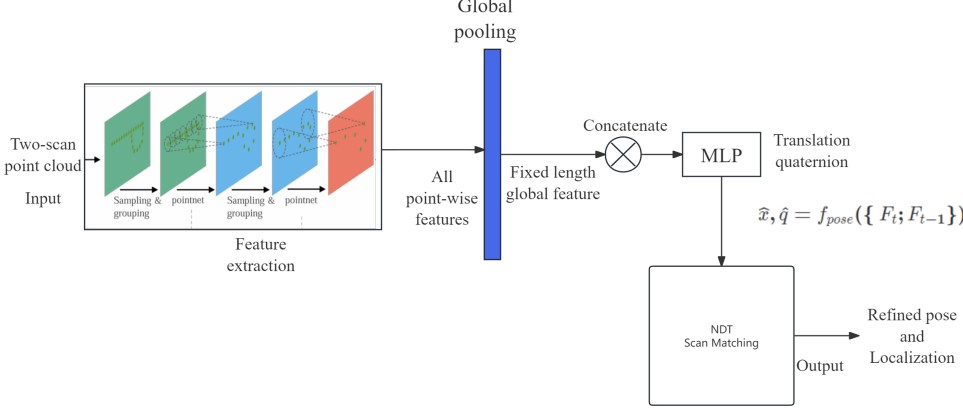

**Figure 1** **General architecture of the proposed solution.**

# RELATED WORK

## Point-based method

Iterative closest point (ICP) (*Ji et al., 2017*) is a commonly used algorithm for matching corresponding points between two point clouds by searching their corresponding relations while minimizing square errors. As this search is based on the gradient descent method, the algorithm requires a good initial value for convergence. *Li et al. (2020)* used ICP for LiDAR scan matching. Their method aimed to remove unnecessary points generated during LiDAR scanning to reduce computation search time when handling large-scale point cloud data. This was accomplished by first extracting the image-based ground points and then segmenting the remaining point cloud into disjoint sets. The six degrees of freedom transformations between consecutive scans were then calculated using point-to-point ICP. While standard point-to-point ICP exhibited better performance in an urban scene, all variants of ICP methods performed poorly in highway scenes. Attempts to improve ICP algorithms have been made (*Li et al., 2014*; *Fieten, Radermacher & Heger, 2012*; *Jinxia & Yuehong, 2011*; *Dai & Yang, 2011*), but improvements are dependent on having sufficient geometric features. *Wan et al. (2018)* highlighted that ICP cannot work in scenes such as highways where there are minimal 3D features available.

## Distribution-based method

Another scan matching algorithm, NDT, is less prone to sensor noise as it does not require corresponding points for point-to-point registration (*Sobreira et al., 2019*). Instead, the scanned point cloud data is matched to a set of normal distributions that is transformed from the reference point cloud (*Biber & Straßer, 2003*).

*Wen, Hsu & Zhang (2018)* proposed using NDT-based graphs to perform SLAM in urban environments. Their method first obtained odometry measurements from the transformation between consecutive scans of point clouds, then built a graph using the obtained measurements. This method then calculated the pose estimate by solving the graph-based optimization problem. Similar to the solution proposed in the present study,

their method focused on using LiDAR for pose estimation, however, the solution proposed in the present study employs deep learning to estimate the initial value for NDT.

While NDT exhibits robustness against noise and has relatively good convergence speed under specific cases (*Sobreira et al., 2019*), related works (*Wang et al., 2020*; *Murakami et al., 2020*; *Magnusson, 2009*; *Magnusson et al., 2009*; *Al-Nadawi et al., 2020*) are inherently sensitive to the initial estimate found in NDT (*Ilci & Toth, 2020*). Therefore, the proposed solution establishes the initial value with a guided estimation approach instead of using a randomly assigned value.

## Feature-based method

LiDAR Odometry and Mapping (LOAM) is a feature-based solution. Geometric features are first extracted from point cloud scans and then used to find the point correspondences between scans (*Ramezani et al., 2020*).

Using the traditional LOAM method, *Zhang & Singh (2014)* implemented real-time mapping without using high-accuracy ranging or inertial measurements by running two algorithms that estimate LiDAR velocity and perform point cloud registration, respectively. *Deschaud (2018)* proposed using 3D depth sensors to achieve low-drift LiDAR odometry and improve map quality by using a scan-to-model matching framework. To handle dynamically changing environments, *Ding et al. (2020)* combined both global matching and LiDAR inertial odometry in a pose graph framework for prompt map updates. For scenes with extreme motion, *Chen et al. (2020)* adopted semantic segmentation and LOAM to estimate tree diameter. Several recent studies (*Li et al., 2019a*; *Jinxia & Yuehong, 2011*; *Ramezani et al., 2020*; *Cho, Kim & Kim (2020)*) indicate that deep learning can be used to improve estimate accuracy by offering automatic parameters that capture geometric features.

Many recent studies have leaned on deep-learning approaches, especially convolutional neural networks (CNN), for more robust feature extraction. *Yin et al. (2018)* proposed LocNet, a pioneer model that leverages point cloud data with CNN for 3D point cloud-based place recognition. Place recognition is a task closely related to LOAM. Registering a frame of a local LiDAR scan into the segmentation of the global HD-map where that frame is likely being scanned can improve LOAM. In LocNet, each frame of a LiDAR scan is divided into groups of horizontal rings. A histogram is then formed with the number of points and their LiDAR intensities falling into each group of rings. The histogram is then rearranged into a 2D image as a reduced representation of each reading; this process is called cylindrical projection (*Chen et al., 2017*). The CNN then takes in a pair of scans that corresponds to either the same or different places, and uses contrastive loss to perform pairwise learning to maximize distance between different places in the metric space. Although this CNN-based place recognition is an automated learning process *via* supervised learning, the rotation invariance is achieved by handcrafting the 2D histogram. The 3D to 2D projection can also lead to information loss. Oreos (*Schaupp et al., 2019*) is a variant of LocNet (*Yin et al., 2018*) for both place recognition (3-DoF registration) and yaw angle estimation, but it also requires 3D to 2D projection.

*Li et al. (2019b)* proposed LO-Net, a deep learning model for 6-DoF pose estimation. LO-Net regresses 6-DoF transformation between consecutive frames of scans and performs point-wise normal estimation and point cloud semantic segmentation. For pose estimation, LO-Net first projects each 3D point onto a 2D plane, and preserves the LiDAR intensity and depth information in multiple channels. Then, a CNN-based Siamese network is trained on the projected scan, supervised with odometry regression loss, as described in *Kendall, Grimes & Cipolla (2015)*.

## Point cloud-based feature learning

Unlike regular structural data, including speeches in linear structure and 2D images in 2D lattice structure, point cloud data is unstructured and has only received attention from researchers in recent years because of the exponential growth seen in computational power. Models have been proposed that effectively use point cloud data alongside successful standard deep learning techniques for normal data formats. PointNet (*Qi et al., 2017a*) is a deep learning network model that directly processes 3D point cloud data. It was initially proposed to address classification and segmentation tasks on point cloud data, applying multi-layer perceptron (MLP) point-by-point to learn permutation invariant features for each point and then performing feature aggregation from all points. PointNet++ (*Qi et al., 2017b*) took PointNet a step further to aggregate neighborhood information instead of only learning point-specific features, improving the discovery of texture clues and reducing the effects of noise.

## METHODS

As mentioned in the previous section, previous 2D image-based approaches (*Li et al., 2019b*; *Schaupp et al., 2019*; *Chen et al., 2017*) suffer from information loss in the cylindrical projection step. Therefore, we propose an end-to-end point cloud-based deep learning network for LiDAR odometry that directly learns features by encoding the structural information of the point cloud scan from the 3D space and then uses the encodings of two frames to estimate their relative positions. The objective of this model is to estimate the 6-DoF transformation parameters between two scans.

Several things were considered in the design of the network architecture. First, in order to make full use of the local regional information adjacent to each point on the scan, the model should be aware of spatial adjacency. Second, point-wise feature extraction should output the same results regardless of the order of the 3D point in the input space. This precludes sequential models like recurrent neural networks (RNNs). RNN is a special neural network structure for processing sequence data, such as natural language or time series. RNNS can require excessive processing time when processing point cloud data and are not intuitive or efficient at extracting spatial features. Third, the scale of translational distance between two input frames of a scan should be confined in a relatively small scope to make regression easier. PointNet can efficiently capture the hierarchical structure and correlations of point cloud data, but has poor local feature extraction capability, which may be limited when dealing with more complex scenarios and requires more computing resources and time when dealing with large-scale or complex point clouds. Because of these

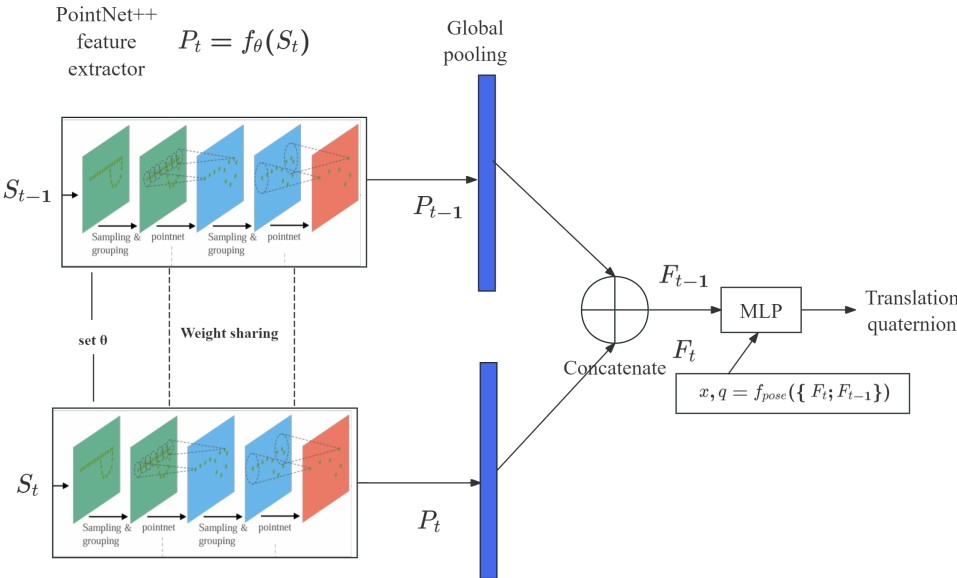

**Figure 2  Proposed pipeline for coarse pose estimation with consecutive LiDAR scans.**

limitations to Pointnet, we used the feature extractor in PointNet++ (*Qi et al., 2017b*) to encode the LiDAR scans and fine-tune the scan-level encodings of both frames to infer the relative 6-DoF position using a set of dense layers. For spatial continuity, we only chose consecutive pairs of frames in the odometry sequences as input scans. To our knowledge, this is the first study to introduce point cloud-based deep learning models to LiDAR odometry.

Our model takes in two consecutive frames of scans ($S_{t-1}$; $S_t$), where each scan $S = \{X^\iota, I^\iota\} N\iota = 0$ contains the 3D coordinates $\chi \epsilon \mathbb{R}^{N \times 3}$ and LiDAR intensities $I \epsilon \mathbb{R}^{N \times 1}$ for all points. The same PointNet++ feature extractor $f\theta : \mathbb{R}^{N \times 4} \rightarrow f\theta : \mathbb{R}^{N' \times K}$ ($N'$ is the point number altered by PointNet++, $K$ is feature dimension) is used with the same parameter set $\theta$ for both scans to obtain the encodings:

$$\boldsymbol{P}_t = f_\theta(\boldsymbol{S}_t), \boldsymbol{P}_{t-1} = f_\theta(\boldsymbol{S}_{t-1}). \tag{1}$$

Then, a global pooling operation is applied to aggregate all point-wise features for both scans $\boldsymbol{P}_t$, $\boldsymbol{P}_{t-1}$ into a fixed length global feature $\boldsymbol{F}_t, \boldsymbol{F}_{t-1} \epsilon R^{1 \times K}$ respectively. Finally, a multi-layer perceptron takes in the global encodings of both scans and outputs pose parameters, including translational pose prediction $\hat{x}$ and 3D rotational pose $\hat{q}$ represented by a quaternion, using the mapping

$$\hat{x}, \hat{q} = \boldsymbol{f}_{pose}(\{\boldsymbol{F}_t; \boldsymbol{F}_{t-1}\}). \tag{2}$$

Because the PointNet++ feature extractor $f_\theta$ concentrates salient features corresponding to key shapes in the point cloud, the scans must be compared using $f_{pose}$, similar to the LO-Net model (*Li et al., 2019b*). The architecture of our proposed model is illustrated in Fig. 2.

## Supervision

Odometry regression loss, originally used in visual odometry tasks, was used to simultaneously learn the parameters of the end-to-end pose estimation network for translations and orientations (*Li et al., 2019b*; *Kendall, Grimes & Cipolla, 2015*). The odometry regression loss calculations are as follows, excerpted for clarity:

$$\mathcal{L}_{pose}(St, St-1) = e^{-\lambda_x}\mathcal{L}_{tr} + e^{-\lambda_q}\mathcal{L}_{rot} + \lambda_x + \lambda_q \tag{3}$$

$$\mathcal{L}_{tr}(S_t, S_{t-1}) = \|\hat{X}_t - X_t\|2 \tag{4}$$

$$\mathcal{L}_{rot}(S_t, S_{t-1}) = \|\hat{q}_t - \frac{q_t}{[[q_t]]1}\|2 \tag{5}$$

where $\lambda_x$ and $\lambda_q$ are learnable loss weights of translational error and rotational error, respectively, which can help stabilize the degrees of penalty applied to them.

## Holistic framework with NDT

Despite the novelty in the design choices of odometry regression loss, it still struggles with reliable pose prediction in empirical circumstances. However, odometry regression loss can serve as a hint provider for UGV localization in mature algorithms (see 'Point-based method'), including NDT. Since NDT is non-deterministic and relatively sensitive to initial conditions, providing a reasonable heuristic for NDT could reduce the iterations required to converge the NDT and improve its accuracy. As shown in Fig. 3, our proposed model uses both odometry regression and point-based NDT in a holistic system, with NDT generating a refined pose for odometry. Figure 4 explains the specific algorithm flow of NDT.

# EXPERIMENTS

## Dataset

The KITTI odometry benchmark dataset (*Geiger, Lenz & Urtasun, 2012*) was used in all the experiments for deep model training as well as the holistic pipeline ('Holistic framework with NDT'). The KITTI odometry dataset provides 11 sequences of empirical urban scenes taken from different locations for training and validation. Each sequence includes LiDAR point clouds, binocular RGB and grayscale images, and frame-by-frame ground truth poses. In these experiments, only the LiDAR point clouds were used in all the training and validation sequences and all other modalities were ignored.

## Training details

Following the dataset split for comparison described by *Li et al. (2019b)*, Sequence 00-06 in the KITTI dataset was used for training and Sequence 07–10 was used for validation. For training sequences, in the preprocessing phase of the point cloud, the size of the grid directly affects the effect and accuracy of downsampling. A smaller grid can retain more details, but has a higher computation time. A larger grid will oversimplify the data of the point cloud, resulting in the loss of important features. After repeated trial and error, a

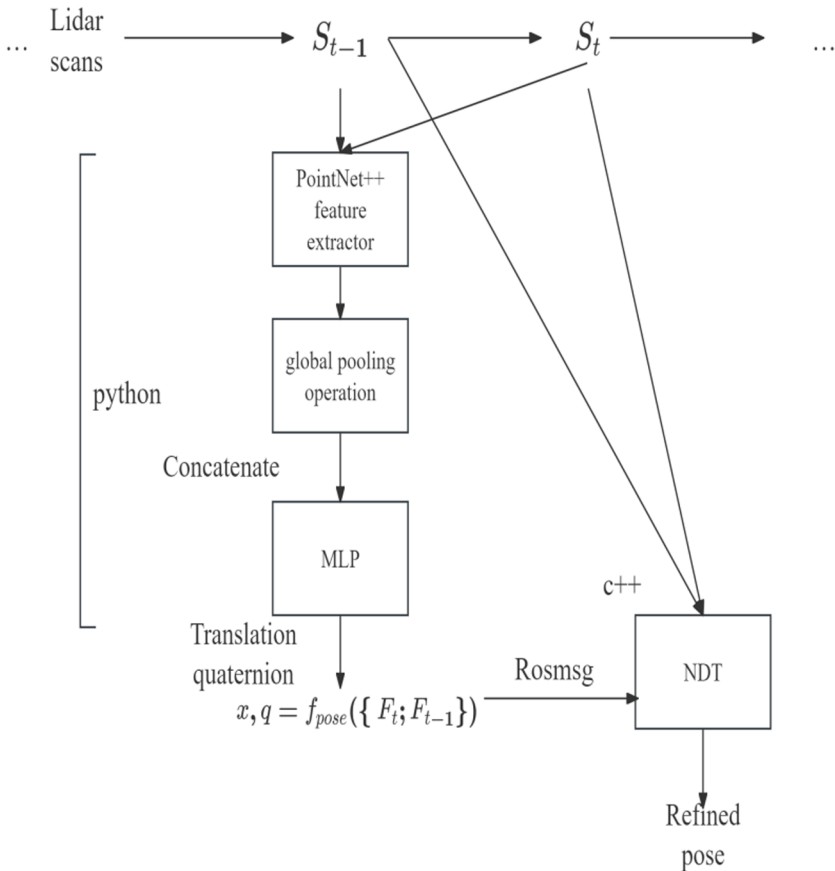

**Figure 3** Implementation of coarse and refine pose estimation cooperation under robot operating system (ROS) system.

voxel grid with a grid size of 3 m in all three dimensions was chosen to first sample each scan. To improve the accuracy and stability of the data, while also decreasing computation time, the scale of the point coordinates were normalized into $[-1, 1]$ intervals in all three axes, and the coordinates of the points were randomly shifted with the maximum translation in each axis set to $1 \times 10-3$. The same data augmentation was also applied to validation sequences, except random coordinate shifting was not performed.

PyTorch[1] was used as the platform for model construction, optimization, and inference, and the PointNet++ module was implemented with the torch-geometric library (*Fey & Lenssen, 2019*). Adjusting the sampling ratio effectively controls the network's ability to capture local and global features of point cloud data at different levels. Radius search determines the size of the local neighborhood of each sampling point and affects how finely the model can capture local features. After repeatedly adjusting the parameters, the final finetuned PointNet++ feature extractor structure parameters were set: the FPS sampling ratio of two SA modules was set to 0.5 and 0.25, and the radius of the radius search was set to 0.2 and 0.4, respectively. The output dimensions in each layer of the point-wise MLP in the first SA module were 1 + 3, 64 and 128, and the output dimensions in each layer of the

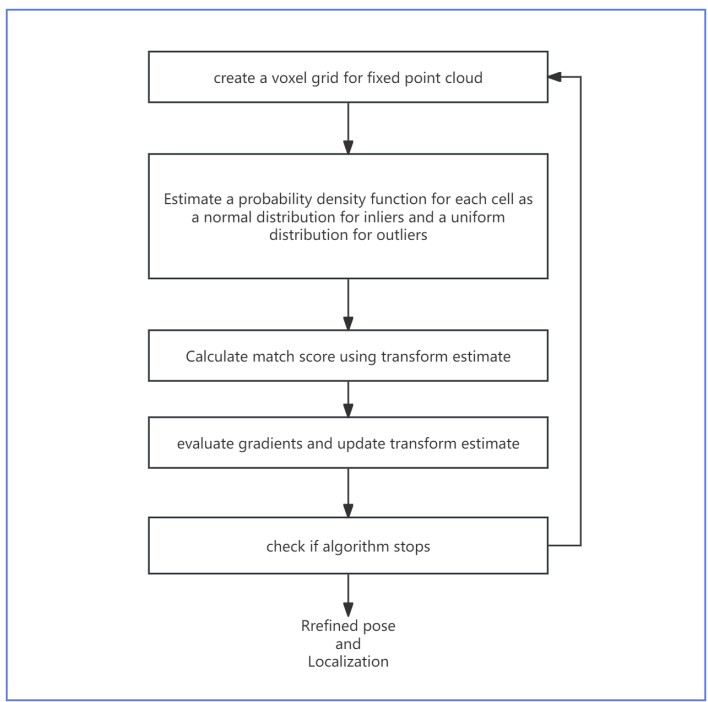

**Figure 4** The specific algorithm flow of NDT.

point-wise MLP in the second SA module were 128 + 3, 128 and 256. In the top regression MLP, the output dimensions of two dense layers were 128 and 64, with a dropout layer after the first layer and a dropout rate of 0.5. The dropout rate of 0.5 effectively reduced overfitting of the model; if the dropout rate is too large or too small, the model cannot learn the features of the data and overfitting can occur. LeakyReLU was used to avoid gradient vanishing when encountering negative activations.

The Adam optimizer was used to train the network with a base learning rate of $10^{-4}$ and reduce the learning rate by a factor of 0.8 on plateau. The minimum learning rate possible was set to $10^{-5}$. The initial values of the two loss weights in Eq. (3) were set to $\lambda_x = -2.5$, $\lambda_q = -2.5$, differing from those used by *Kendall, Grimes & Cipolla (2015)* as the translations in our specific dataset were more diverse than rotational datasets, and thus harder to train. After a few training iterations, the two loss weights stagnated at around 1.5 and 1.8, respectively. To further mitigate overfitting in training, the L2-norm of all trainable parameters (loss weights not included) was introduced as the regularization term in total loss function, with regularization term weight set to $5 \times 10^{-3}$. The model was trained for over 50 epochs with a batch size of three on a GTX1060 GPU personal laptop with 6 GB of VRAM.

## NDT settings

The C++ implementation of NDT in Point Cloud Library (PCL)[2] was used in this model. Before NDT started to align the input point cloud $S_t$ to the target point cloud $S_{t-1}$, a Kd-Tree was constructed using the target point cloud. To speed up the internal searching

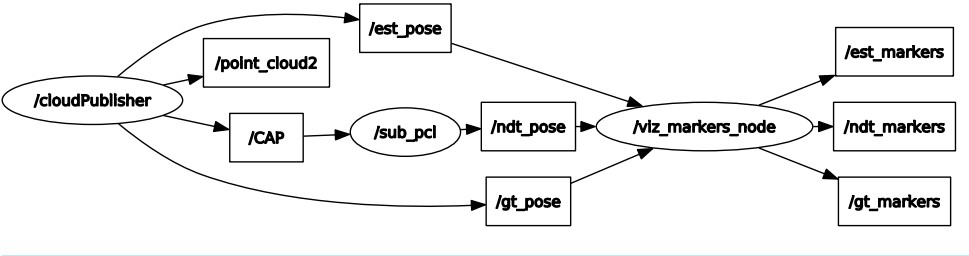

**Figure 5    ROS graph.**

of NDT without losing too much precision, the input point cloud was downsampled with a voxel grid size of 0.2 m in all three dimensions.

## ROS system settings

Since the deep regression model was implemented in Python while NDT was implemented in C++, both models were deployed in two robot operating system (ROS) nodes and ROS topic was used to communicate between them. ROS Melodic[3] under Ubuntu 18.04 was used for ROS distro. An ROS system was constructed, as shown in Fig. 5, where */cloud Publisher* was the point cloud IO and deep model inference node, */sub_pcl* was the NDT processing node and /CAP was the customized ROS message encapsulating *sensor_msgs/PointCloud2 and geometry_msgs/TransformStamped messages.* These structures were set up for easy communication between individual nodes. In the deep learning model, the rough estimated position was */est_pose, /ndt_pose* was the position after the NDT scan matching algorithm, and */gt_pose* was the real position. The final node was the *vtz_markers_node*, which displayed the rough estimated position, the NDT scan matched position, and the real position.

## Quantitative results

We compared the translational and rotational errors of our methods with baseline methods on each sequence. As shown in Table 1, for translational measurements, the proposed deep pose coarse estimation performed better than the LO-Net with mapping method on four out of seven training sequences and all validation sequences, reflecting overall stronger average scores for all corresponding sets of sequences. For rotational error, only the NDT method with customized initialization slightly outperformed the best baseline method. In terms of runtime performance, as shown in Table 2, the holistic pipeline of the proposed model was able to reduce the total time of localization. The NDT algorithm converged faster when taking the coarse pose from the deep model, and the total runtime for each frame also dropped around 1.32 s on average. Since the runtime test was performed inside Virtual Machine, the speed improvement would be more pronounced if running on a full-stack, physical machine.

Despite the promising precision scorings, the translational RMSE measure did not completely reflect the compliance of the whole odometry track to the ground truth. Figure 6 shows that our model generated volatile predictions in a large time span, with *gt* representing the real trajectory and *est* representing the trajectory output by the proposed model. In the beginning, the error between the output trajectory of the proposed model

**Table 1** Transition and rotational RMSE comparison between the proposed approach and baseline.

| Methods | Metrics | Training sequences | | | | | | | Validation sequences | | | | Train mean | Validation mean |
|---|---|---|---|---|---|---|---|---|---|---|---|---|---|---|
| | | 0 | 1 | 2 | 3 | 4 | 5 | 6 | 7 | 8 | 9 | 10 | | |
| LO-Net (*Li et al., 2019b*) | TL | 1.47 | 1.36 | 1.52 | 1.03 | 0.51 | 1.04 | 0.71 | 1.70 | 2.12 | 1.37 | 1.80 | 1.09 | 1.75 |
| LO-Net + mapping (*Li et al., 2019b*) | TL | 0.78 | 1.42 | 1.01 | 0.73 | 0.56 | 0.62 | 0.55 | 0.56 | 1.08 | 0.77 | 0.92 | 0.81 | 0.83 |
| PN++ only | TL | 0.64 | 0.92 | 0.77 | 0.81 | 0.85 | 0.59 | 0.58 | 0.56 | 0.72 | 0.75 | 0.66 | 0.76 | 0.67 |
| NDT | TL | 1.11 | 1.28 | 1.16 | 1.01 | 1.08 | 1.09 | 0.96 | 1.04 | 1.18 | 1.01 | 1.26 | 1.09 | 1.12 |
| PN++ + NDT | TL | 1.07 | 1.28 | 1.24 | 1.01 | 1.08 | 1.09 | 0.94 | 0.92 | 1.12 | 1.23 | 1.16 | 1.11 | 1.11 |
| LO-Net (*Li et al., 2019b*) | RL | 0.72 | 0.47 | 0.71 | 0.66 | 0.65 | 0.69 | 0.50 | 0.89 | 0.77 | 0.58 | 0.93 | 0.63 | 0.80 |
| LO-Net + mapping (*Li et al., 2019b*) | RL | 0.42 | 0.40 | 0.45 | 0.59 | 0.54 | 0.35 | 0.33 | 0.45 | 0.43 | 0.38 | 0.41 | 0.44 | 0.42 |
| PN++ only | RL | 0.67 | 0.64 | 0.68 | 0.5 | 0.29 | 0.64 | 0.64 | 0.85 | 0.7 | 0.7 | 0.69 | 0.58 | 0.74 |
| NDT | RL | 0.61 | 0.21 | 0.55 | 0.37 | 0.2 | 0.54 | 0.53 | 0.86 | 0.57 | 0.55 | 0.43 | 0.43 | 0.60 |
| PN++ + NDT | RL | 0.51 | 0.21 | 0.55 | 0.37 | 0.2 | 0.54 | 0.51 | 0.84 | 0.57 | 0.41 | 0.41 | 0.42 | 0.57 |

**Notes.**

Abbreviations: tl and rl, stand for translational RMSE score (%) and rotational RMSE score (deg/100 m), respectively, using the KITTI odometry benchmark; pn++, the results of the proposed point cloud-based coarse pose estimation model; NDT, the results using NDT with the default initial guess; pn++ + ndt indicates the results using NDT with coarse estimation as the initial guess.

**Table 2  Average required frame-based running time for all frames in sequence 10 in KITTI dataset.**

| Approach | Inference [s] | NDT [s] | Total [s] |
|---|---|---|---|
| w/o PN++ | – | 2.77 | 2.77 |
| with PN++ | 0.352 | 1.096 | 1.448 |

Notes.
(pn++ stands for for our point cloud based coarse pose estimation model).

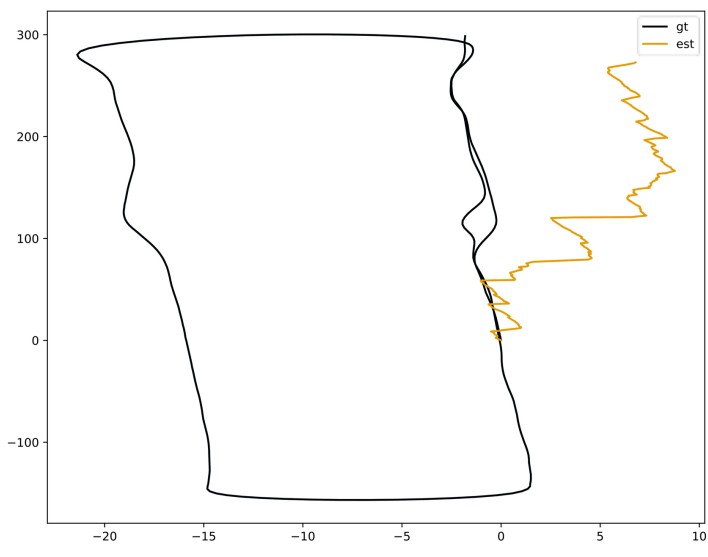

**Figure 6  Bird's-eye view of odometry trajectories of Sequence 06 comparing ground truth and coarse pose estimation.**

and the real trajectory was relatively large, but after a period of time, these values converged with the output trajectory of the proposed model gradually approaching the true trajectory. This indicates that as the data increases, the estimated position output by the proposed model gets closer to the true estimate.

# REFLECTIONS

## Limitations of PointNet++ finetuned model

The hyper parameter settings used in this model were by no means the optimal selections, but were an initial attempt to introduce emerging point cloud-based deep learning techniques. As shown in the above section ('Quantitative results'), the proposed method still faces challenges in terms of prediction diversity and robustness compared to de facto image-based methods. To fully demonstrate the benefit of leveraging LiDAR data with point cloud-based deep learning, the limitations of the proposed model were analyzed and methods of mitigating these limitation were proposed:

### Imbalanced dataset

In the ground truth trajectories in the dataset recorded in well-developed urban traffic, the frames with straightforward transition and small rotation overpowered the poses generated

by turning around, making it hard for the model to fit such imbalanced prior distribution. Future works should apply a hard-negative mining training strategy to mitigate this imbalance problem.

### Limited scene and perspective diversity of the dataset

The dataset we used mainly contains point cloud data of urban driving scenes, and the perspective is relatively fixed, usually from the top-down perspective of the vehicle. This can lead to limitations in the PointNet++ network's ability to generalize when dealing with other scenarios or point clouds from different perspectives. For tasks that need to deal with a wider range of scenarios and perspectives, more diverse training data may be required.

### Little constraints on relevant pose feature extraction

Although point cloud-based methods prevent information loss during the preprocessing, they may not be able to prevent information loss in feature extraction. The proposed model used PointNet++ to obtain encoding in parallel for two input scans, without interactions in between. The underlying assumption was that the feature extractor would be competent enough to generate a discriminative global encoding of the scan for downstream odometry regression. However, this assumption may not be true with large-variance empirical data, where standard point-wise methods stated in 'Point-based method' are more prevalent. Sharing intermediate features between feature extractors, especially shallow point-wise or local features, is likely to help extract more relevant pose features.

### PointNet++ is sensitive to point density during training

Pointnet ++ treats each point as a separate entity, regardless of its density or distribution. This can lead to performance degradation when dealing with non-uniform or sparse point clouds. In these cases, the model may have difficulty learning to distinguish features effectively, resulting in reduced accuracy. To improve performance with non-uniform or sparse point clouds, the grid size can be dynamically adjusted, and the number of sampling points can be reduced in a dense area and increased in a sparse area. This would allow the model to focus evenly on regions of different densities.

## Limitations of current project

Aside from the limitations mentioned in the previous section, the proposed model's cooperation with NDT also has potential for further improvements. The current standard NDT algorithm performs neighbor point searching using a Kd-Tree in the original 3D space, but the results of PointNet++ with NDT could be improved by employing an alternative searching algorithm that can use high-level features as fingerprints. This could further boost searching efficiency within NDT.

## Enhancing the current search method

The vector quantization (VQ) (*Gray & Neuhoff, 1998*; *Gray, 1984*) method is an example of a searching algorithm that could improve the results of the proposed model as VQ can effectively search in high-dimensional embedding space. Product quantization (PQ) (*Douze, Jegou & Schmid, 2011*) is a VQ method particularly good at handling a large number of code words. The key idea of PQ is to decompose the original vector space into

the cartesian product of $M$ low-dimensional sub-spaces and quantize each subspace into $k$ code words. The effective number of code words in the original space is $kM$, but the cost of storing them is merely $O(Dk)$ for D-dimensional vectors, making it relatively space efficient.

PQ has been used for compact encoding and approximate distance computation in nearest neighbor searches, and has recently been used in combination with the inverted indexing method (*Babenko & Lempitsky, 2012*) to prevent exhaustive searches and speed up the search process. Given the time complexity of the $O(Mk)$ per distance computation using lookup tables for PQ, the methods presented by *Douze, Jegou & Schmid (2011)* and *Babenko & Lempitsky (2012)* are state-of-the-art methods for compact encoding and inverted indexing, respectively. Indexing and searching the database using the above method could be concisely described as follows:

**Indexing** a vector $y$:

1. Quantize $y$ to $q_c(y)$
2. Calculate residual $r(y) = y - q_c(y)$
3. Quantize $r(y)$ to $q_p(r(y))$, which for the product quantizer is equivalent to assigning $u_j(y)$ to $q_j(u_j(y))$, for $j = 1...m$.
4. Add a new entry to the inverted list corresponding to $q_c(y)$ containing the vector (or image) identifier and the binary code (the product quantizer's indexes).

**Searching** the nearest neighbor(s) of a query $x$:

1. Quantize $x$ to its nearest neighbors ($w$) in the codebook $q_c$; $r(x)$ denotes the residuals associated with these $w$ assignments. The following two steps are applied to all $w$ assignments:
2. Compute the squared distance $d(u_j(r(x)), c_{j,i})^2$ for each subquantizer $j$ and each of its centroids $c_{j,i}$;
3. Calculate the squared distance between $r(x)$ and all the indexed vectors of the inverted list. Using the subvector-to-centroid distances computed in the previous step, the sum of the $m$ look-up values is calculated and then the $K$ nearest neighbors of $x$ are selected based on the estimated distances. This is implemented efficiently by maintaining a Maxheap structure of fixed capacity that stores the smallest $K$ values seen so far. After each distance calculation, the point identifier is added to the structure only if its distance is below the largest distance in the Maxheap. where $q_c$ is a quantizer learned using k-means, referred to as the coarse quantizer; $q_p$ is the product quantizer used to encode the residual vector; and $u_j$ is the $j$th subvector of vector $x$.

### Enhancement experiments and result

We conducted the experiments on both Autoware (*Molina et al., 2017*) and Baidu Apollo (*Wan et al., 2018*) platforms on a local PC. These platforms are currently the only two well-known open-source platforms for the whole lifecycle in autonomous vehicle development. We modified the NDT implementations in both platforms, using PQ to replace the default Kd-Tree-based searching method, and used the point cloud datasets provided by the platforms for verification. For Autoware, the dataset was found on page 4 of its project wiki. For Baidu Apollo, we downloaded source code5 and used the dataset under the

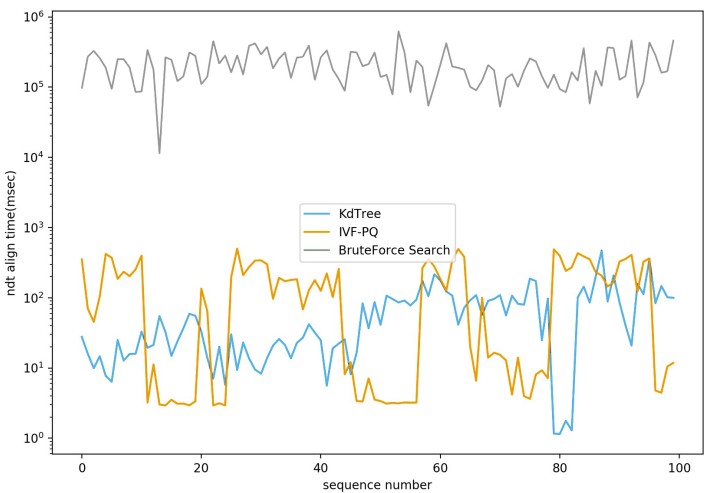

**Figure 7** **NDT align time comparison based on different nearest neighbor search algorithms.**

NDT module for verification. As shown in Fig. 7, we compared the searching time of the PQ method with those of the Kd-Tree method and brute-force method on the Autoware dataset of three-dimensional point cloud data. Notably, the results suggested that PQ performed slightly worse than the Kd-Tree method, but still vastly better than brute-force method. However, PQ's search speed would likely be significantly shorter than the Kd-tree method if tested on point cloud datasets with high-dimensional features (*e.g.*, around 1,024 dimensions). As the data dimension increases, the performance of the Kd-Tree tree method decreases sharply. In a high-dimensional space, the data points are very sparse, making the distance calculation inaccurate and resulting in poor partitioning of the Kd-Tree tree. The "dimensional curse" problem in high-dimensional data also makes the Kd-Tree method less efficient because as the dimension increases, the distribution of data points on each node becomes more uniform, causing the depth of the Kd-Tree tree to increase and the search efficiency to decrease. In large-scale datasets, the construction and search process of a Kd-tree requires a lot of computing resources. First, building a Kd-tree involves the recursive partitioning of data, which requires a lot of computation time and memory. Second, in the search process, the distance between the query point and each node in the Kd-tree needs to be calculated, and many comparisons and judgments are made, which also increases the number of calculations. Third, for large datasets, a Kd-tree uses a lot of memory to store data structures and intermediate results. As data size increases, memory consumption can also grow rapidly, which can lead to insufficient computational resources or performance bottlenecks (*Sariel, Indyk & Motwani, 1998*; *Muja & Lowe, 2014*). In contrast, when dealing with high-dimensional large-scale data, the PQ algorithm can decompose the original high-dimensional space into Cartesian products of several low-dimensional vector spaces and quantize them individually, significantly reducing memory storage requirements and computational complexity and improving computational efficiency.

**Future work**

Based on the results of the enhancement studies, the following steps could also improve the proposed model:

1. Modifying the PCL in the NDT module code so that it can support the processing of high-dimensional point cloud data.

2. Adaptively adjusting the radius search value. LiDAR points are traditionally retrieved by finding k nearest neighbors or all points included in a small confined environment centered on the point of interest. However, k and ambient radius values are usually selected heuristically and are assumed to be constant for the entire point cloud, rather than guided by data. This does not ensure that all of the identified neighbors belong to the same object as the current point. Therefore, when several different structures are included, their local descriptions may be biased and provide the wrong feature descriptors.

3. Because the three main approximate nearest neighbor (ANN) methods have different adaptation scenarios, these three main methods could be implemented in the NDT module then the method to run could be chosen dynamically based on the surrounding environment at runtime.

## CONCLUSION

Driving an autonomous vehicle safely requires the vehicle to perceive its surroundings and localize itself accurately so that sound and agile decisions can be made in real world environments. In GPS-denied locations, autonomous vehicles need an alternate sensor. This article focuses on LiDAR as the primary sensor for simultaneous localization and mapping. To improve pose estimate accuracy, PointNet++ was implemented to extract geometric features in point cloud data and output the initial pose estimate. The localization output was then generated by initializing NDT with the initial pose estimate. The system architecture of the proposed model comprises a deep learning framework and scan registration method for localization. The proposed model was then tested using a ROS and the estimated pose results were compared against the ground truth using the RVIZ visualizer. In the attempt to improve internal neighborhood search in NDT, the PQ method was also explored to maximize the high-dimensional features of PointNet++.

### Funding

The authors received no funding for this work.

### Competing Interests

The authors declare there are no competing interests.

## Author Contributions

- Wang QiXin conceived and designed the experiments, performed the experiments, analyzed the data, performed the computation work, prepared figures and/or tables, authored or reviewed drafts of the article, and approved the final draft.
- Wang Mingju analyzed the data, prepared figures and/or tables, and approved the final draft.

## Data Availability

The code is available at Zenodo: Wang, Q. (2024). Kitti_odom. Zenodo. https://doi.org/10.5281/zenodo.11398595

The KITTI Vision Benchmark Suite dataset is available at: https://www.cvlibs.net/datasets/kitti/raw_data.php.

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
