# Peer review of "A novel 3D LiDAR deep learning approach for uncrewed vehicle odometry"

_PeerJ Computer Science, doi:10.7717/peerj-cs.2189_

## Round 0.1 · original submission · Major Revisions

Dear authors,
You are advised to critically respond to all comments point by point when preparing a new version of the manuscript and while preparing for the rebuttal letter. Please address all the comments/suggestions provided by the reviewers.

Kind regards,
PCoelho

**Language Note:** The review process has identified that the English language must be improved. PeerJ can provide language editing services - please contact us at [email protected] for pricing (be sure to provide your manuscript number and title). Alternatively, you should make your own arrangements to improve the language quality and provide details in your response letter. – PeerJ Staff

Reviewer 1 ·

Basic reporting

The paper is articulated in clear, professional English, though minor grammatical adjustments could further improve readability. It sets a strong context for the necessity of precise localization in autonomous vehicles, with a thorough literature review that adeptly identifies gaps this study seeks to address.

The structure adheres to academic standards; however, it would benefit from the inclusion of an overall architecture representation and a dedicated contributions section.

Concerning figures, there are areas for improvement:

Figure 1 seems unnecessary as the SAE levels of driving automation are well-known.
Figure 2 representation is overly simplistic, lacking detailed inner modules and inputs and outputs.
Figures 3 and 4 should align their terminologies more closely with the text for coherence.
Figure 5 appears to contribute little to understanding the method.
Figure 6 is not effectively described in the text, making its interpretation challenging.
Figure 7 would benefit from further discussion within the text.

For Table 1, consolidating TR and TL results into a single table or separating them distinctly could enhance clarity. Highlighting the best results would also provide a clear indication of the study's key findings.

Several pseudocode algorithms representation would clear the readers understanding of the proposed method.

Experimental design

The research presented is original and aligns well with the journal's focus, addressing a notable gap in LiDAR-based odometry for autonomous vehicles. A more in-depth discussion on the limitations and potential biases in the dataset and model training would strengthen the paper, ensuring the findings' robustness and generalizability.

The paper articulates the research question effectively, demonstrating how the proposed method bridges the identified knowledge gap. It innovatively combines deep learning with traditional algorithms to refine pose estimation accuracy.

The methods section is sufficiently detailed, facilitating replication.

Using the KITTI dataset for quantitative results boosts the study's trustworthiness.

Although the results are modest, the analysis of limitations is insightful, suggesting pathways for significant future improvements

Validity of the findings

The detailed description of methods, from the use of PointNet++ for feature extraction to the integration of Product Quantization with NDT, shows a thorough approach to solving the problem. To further strengthen the paper, the authors could consider a more detailed discussion on the choice of deep learning architecture and its implications on the results.

The paper provides a thorough analysis of the collected data, ensuring robust and statistically sound findings. However, further details on data preprocessing and parameter selection could enhance replicability. The paper major strength lies in its innovative combination of deep learning with LiDAR odometry, promising improvements in localization accuracy.

The findings offer a significant advancement by integrating Product Quantization with NDT, improving the neighborhood search in high-dimensional feature spaces. However, the comparison with existing methods revealed modest results and the discussion on the implications on improvements is not suficient to account as a contribution of this research work.

The conclusions are well-articulated, linking back to the research question and supported by the results. The discussion on limitations and future work adds depth to the study, acknowledging areas for improvement.

Cite this review as

Reviewer 2 ·

Basic reporting

Clarity and Language: The manuscript is generally well-written with professional language that makes the technical content accessible. However, minor grammatical errors and awkward phrasing in some sections should be revised for better clarity (e.g., lines 12, 49).

Structure: The document is structured appropriately in accordance with PeerJ standards, including clear sections for Introduction, Methods, Experiments, Results, and Discussion.

Literature: The references are well-cited and relevant, providing a robust background for the study's methodologies and its positioning within existing research.

Experimental design

Research Originality and Design: The study's design is sound, focusing on a novel integration of PointNet++ deep learning models with Normal Distributions Transform (NDT) for improving odometry in uncrewed ground vehicles—a well-defined and relevant research gap.

Methods Detail: The manuscript successfully describes the methods with sufficient detail to allow reproducibility, which includes data processing, model training specifics, and integration with NDT algorithms.

Validity of the findings

Data Robustness and Statistical Analysis: The research utilizes the KITTI odometry benchmark dataset, which is a standard in the field, ensuring that the results are robust and comparable to other studies. The statistical analysis is sound, with appropriate metrics used for evaluating model performance.

Findings and Conclusion: The conclusions are well-supported by the data presented, linking back to the initial research question regarding the effectiveness of the proposed LiDAR feature learning model integrated with NDT. The manuscript also discusses limitations and future work, which shows a good understanding of the study's scope.

Additional comments

The use of a point cloud-based deep learning approach addresses the common issue of information loss in 3D to 2D projections found in other models. The manuscript does well to explain the potential impact and improvements introduced by this method.

However, the discussion on potential enhancements and the exploration of high-dimensional feature searches (Product Quantization) are speculative and not fully backed by experimental results. This part of the study could benefit from additional empirical support.

Cite this review as

---

## Round 0.2 · accepted · Accept

Dear authors,
It was not possible to obtain a new review from reviewer #1 from the previous round, so I verified your submission myself. In this sense, I thank you for the improvements you made to the document, which is why I believe it is ready to be accepted for publication.
Thank you for considering PeerJ Computer Science and submitting your work.

Reviewer 2 ·

Basic reporting

The manuscript is improved.

Experimental design

The manuscript is improved.

Validity of the findings

The manuscript is improved.

Additional comments

The manuscript is improved.

Cite this review as